# Glycerin-Grafted Starch as Corrosion Inhibitor of C-Mn Steel in 1 M HCl solution

**Sihem Lahrour [1], Abderrahim Benmoussat [2],* , Brahim Bouras [1], Asma Mansri [1], Lahcene Tannouga [1] and Stefania Marzorati [3]**

[1] Research Laboratory of the Organic Electrolytes and Polyelectrolytes Application (LAEPO), Department of Chemistry Tlemcen, Faculty of Sciences, University of Tlemcen, Tlemcen 13000, Algeria; lahrour91sihem@gmail.com (S.L.); bourasbrahim_m8@yahoo.fr (B.B.); asma.mansri@yahoo.fr (A.M.); l14_ten@yahoo.fr (L.T.)

[2] LAEPO Research Laboratory, Matter Sciences Department, Institute of Sciences and Technology, CUTAM University Center of Tamanrasset, Tamanrasset 11000, Algeria

[3] Department of Environmental Science and Policy, Università degli Studi di Milano, 7-20122 Milano, Italy; stefania.marzorati@unimi.it

* Correspondence: abbenmoussa@gmail.com

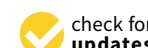

**Featured Application: In the last decade, many research works were focused on the use of "green" corrosion inhibitors, lacking in the adverse health issues connected to the organic compounds used in the past. The use of biomaterials as corrosion inhibitors takes considerable attention due to their inherent stability, availability, cost effectiveness, and environmentally-friendly nature.**

**Abstract:** C-Mn steels, commonly employed in structural applications, are often exposed to near-neutral aerated environments and hence subjected to general corrosion. In broader contexts, for example during pickling, acidizing treatments, or acid-releasing processes, where steel comes in contact with more aggressive solutions, the use of corrosion inhibitors is a supplementary strategy to cathodic protection and/or coating. This work focuses on the C-Mn steel corrosion protection in the presence of HCl, either as process fluid or by product. In order to avoid the toxicological issues related to conventional synthetic products, a bio-copolymer containing glycerin-grafted starch, synthesized by modification of maize starch, was studied as a "green" corrosion inhibitor by the weight loss method and electrochemical techniques (open circuit potential, potentiodynamic polarization and electrochemical impedance spectroscopy). Corrosion-related parameters, such as inhibitor concentration and temperature, were varied and optimized to characterize the corrosion process. Results showed that inhibition efficiency increases with increasing bio-copolymer concentration, reaching a maximum of 94%at the concentration of 300 mg L$^{-1}$. The kinetic and thermodynamic parameters were determined and discussed. The obtained values of corrosion potential and corrosion current density, $E_{corr}$ and $i_{corr}$, obtained by potentiodynamic polarization, are in agreement with the weight loss method. The corrosion current densities decrease when the concentration of the inhibitor increases.

**Keywords:** C-Mnsteel; corrosion inhibitors; bio-copolymer; starch; glycerin

---

## 1. Introduction

Hypoeutectoid C-Mn steels are typically employed in many structural applications and, being commonly exposed to near-neutral aerated environment, undergo general corrosion in the form of rust. Cathodic protection and/or coating, together with the use of adequate corrosion allowance, enables the phenomenon to be controlled, thus reducing the risk of failure [1]. The use of C-Mn steel

in a broader context, where the steel comes in contact with more aggressive solutions (for example, during pickling or oil well acidizing treatments as well as acid-releasing processes [2,3], might need diverse technologies, such as addition of inhibitor products.

In this work, we propose to investigate a method to control and minimize the steel corrosion in the presence of HCl by adding a corrosion inhibitor of bio-polymeric nature. The test solutions are meant to mimic the aggressiveness of the environment due to in-situ generation of harsh byproducts (i.e., during plastic or biomass combustion) or due to an environment that is aggressive itself in the exercise conditions (i.e., pickling or acidizing treatments).

The corrosion inhibition strategy is well-known to help in reducing the economic input of corrosion damage [4,5]. Moreover, through a proper selection of the nature of the used inhibitor compounds, the related toxicological aspects can be controlled. Efficient inhibitors are compounds whose activity is related to the presence in their structure of hetero atoms like nitrogen, oxygen, or sulfur [6]. It was observed that adsorption depends mainly on the presence of lone pair electrons, electron-donating groups and π-orbital character of the molecule [7]. However, even if many synthetic organic compounds are effective inhibitors, regarding their cost and toxic nature, researches were compelled [8,9]. The concern is not only related to the environmental poisoning, needing effective procedures of inhibitor removal before flowing the used solution out of the industrial plant, but also to the contact of the protected metal, for example, with food, beverages, and medicals, during its lifecycle. In the last decade, many research works were focused on the use of "green" corrosion inhibitors, lacking in the adverse health issues connected to the organic compounds used in the past [8,9]. The use of biomaterials as corrosion inhibitors takes considerable attention due to their inherent stability, availability, cost effectiveness, and environmentally-friendly nature [10,11].

The protection of a steel structure against corrosion has already been studied by means of biopolymers as corrosion inhibitors [12,13]. Bio-copolymers derived from starch, already used in the literature as a corrosion inhibitor, motivated our research [14,15].

The present work reports on the establishment of different possibilities of grafting glycerin on starch. We looked at the different grafting sites on amylopectin that can receive glycerin molecules. By the conditions, this was possible by eliminating amylose chains. The obtained product was used as corrosion inhibitor of C-Mn steel in 1 M HCl solution in the temperature range 25–50 °C. The corrosion inhibition was investigated by weight loss measurements, potentiodynamic polarization, and electrochemical impedance spectroscopy (EIS). Appropriate corrosion parameters such as corrosion current density ($i_{corr}$), polarization resistance ($R_p$), and parameters derived from EIS were derived to characterize the corrosion process. The kinetic and thermodynamic parameters were determined and discussed.

## 2. Materials and Methods

### 2.1. Reagents

The used reagents were maize starch (St), glycerin (Gly), hydrochloric acid 37%, and sodium hydroxide. They were supplied by the Aldrich Company. The only solvent used was bi-distilled water.

### 2.2. C-Mn Steel

Tests were conducted on C-Mn steel samples obtained and treated according to the procedure described in our previous work [4].

Atomic composition was compared to the one quoted in the material test certificates [16]. Elemental composition of the cutting samples (Table 1) was determined by emission spectroscopy analysis type "Spectro RP 212". Composition and microstructure can vary significantly, resulting in substantial differences in corrosion performances in a corrosion regime. Chemical composition of steel, displayed in Table 1, was in compliance with the API norm (American Petroleum Institute).

Table 1. Chemical composition of C-Mn steel.

| Element | C | Si | P | Mn | S | Al | Mo |
|---------|------|------|------|------|------|------|------|
| Mass % | 0.129 | 0.290 | 0.016 | 1.590 | 0.018 | 0.024 | 0.008 |
| **Element** | **Cr** | **Cu** | **Pb** | **Zn** | **Ni** | **V** | **Fe** |
| Mass % | 0.015 | 0.024 | 0.0016 | 0.003 | 0.007 | 0.004 | 98.06 |

We observed a low carbon composition (0.129%), providing high chemical resistance, high concentration of manganese (1.590%), and low sulfur and phosphorus content. This composition can lead to the formation of manganese sulfide (MnS) inclusions, which are not desirable in the microstructure, being able to start corrosion pitting [7,17].

The surfaces of steel samples were observed under an optical microscope and SEM (Figure 1). The metallographic images revealed a fine microstructure of ferrito-perlitictype, with ferritic prevalencein the presence of clusters of pearlite in the grain boundaries withsome inclusion fields.

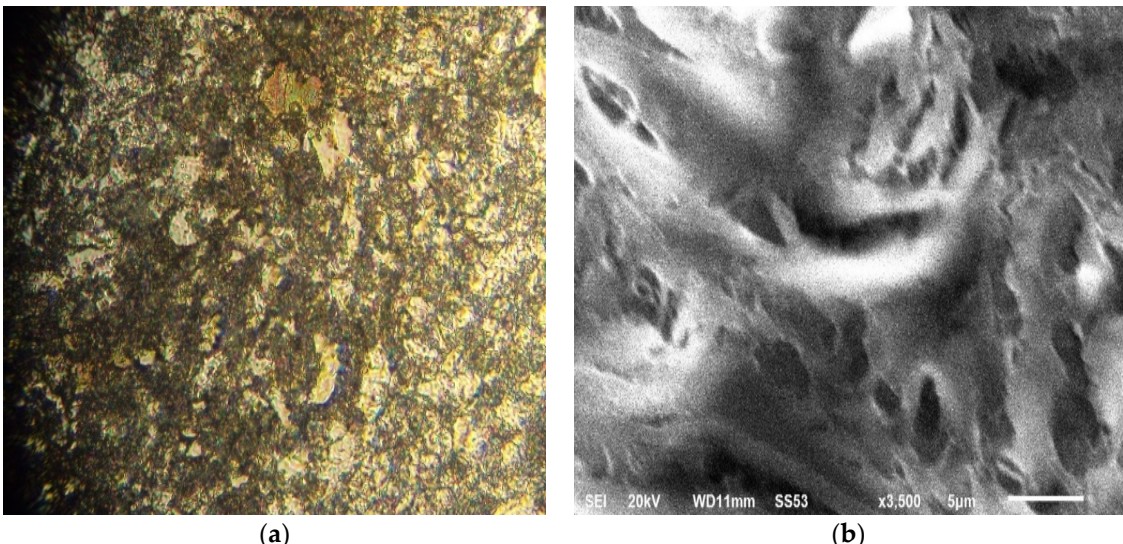

(**a**)　　　　　　　　　　　　　　　　　(**b**)

**Figure 1.** Micrographs of ×60 steel. (**a**) Optical microscope observation; (**b**) SEM imaging (scale bar = 5 μm).

The refinement of ferritic size was obtained by different hardening and precipitation mechanisms based on a dislocation movement that increases elasticity limit and steel resistance. The SEM image (Figure 1b) confirms the fine microstructure of steel.

### 2.3. Test Solution

The chosen solution for corrosion tests was 1 M HCl solution prepared from analytical grade 37% (Aldrich) by dilution with bi-distilled water.

### 2.4. Synthesis of Bio-Copolymer (St63Gly37)

Anew glycerin-starch bio-copolymer (abbreviated from now on by (St63Gly37) was synthesized by modification of maize starch and used as corrosion inhibitor. Glycerin grafting on the starch takes place in three stages. The first and second steps consisted of the preparation of aqueous solutions of 50% glycerin (A) and 50% starch (B), respectively. In the third step, the glycerin (A) and starch (B) solutions were poured into a 200 mL Erlenmeyer flask, to which 3 mL of 1 M HCl were added under stirring and heating to evaporate all the water. Heating and stirring were continued for the sole purpose of evaporating all the water contained in the solution. The obtained mixture was neutralized

by adding 3 mL of 1 M NaOH. The molecular structures of the involved species are displayed in Figure 2.

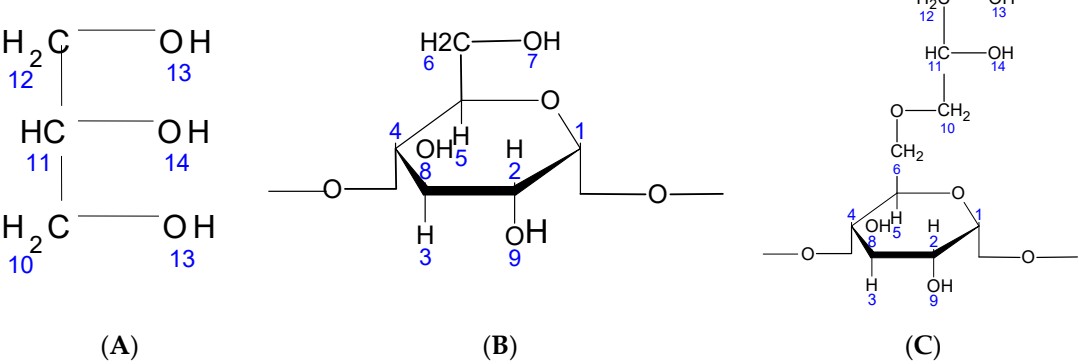

**Figure 2.** (**A**)Chemical structure of glycerin; (**B**) chemical structure of native starch; (**C**) chemical structure of St63Gly37.

### 2.5. $^1$H NMR Characterization of St63Gly37

The $^1$H NMR spectrum of the starch grafted with glycerin St63Gly37 was recorded on a Brucker spectrometer of 400 MHz and dissolved in DMSO-$d_6$ as solvent. The interpretation of the $^1$H NMR results of the product St63Gly37 was carried out by comparison with the peaks of the native starch found in the literature [18,19].

### 2.6. FT-IR Characterization of St63Gly37 Bio-Copolymer

FT-IR spectra were recorded on a Fourier transform infrared (FT-IR) spectrometer Agilent Technologies Cary 600 Series with a resolution of 4 cm$^{-1}$ in the LAEPO laboratory, University of Tlemcen. They were employed to observe the characteristic transmittance bands in St63Gly37. The FT-IR spectra were recorded over the frequency range between 500 and 4000 cm$^{-1}$.

### 2.7. Corrosion Inhibitor Tests

The weight loss (2 cm$^2$ apparent surface area) and electrochemical tests (1 cm$^2$ exposed surface to the corrosive solution) were carried out according to procedures described in previous works [4,13]. St63Gly37 testing was carried out in 1 M HCl solution, varying the concentration of the inhibitor between 5 and 300 mg L$^{-1}$. The temperature range was 25–50 °C.

The inhibition efficiency ($E_w$ %) was calculated using the following Equation (1):

$$E_w = (1 - \frac{W_{corr}}{W^{\circ}_{corr}}) \times 100,$$

(1)

where $W_{corr}$ and $W^{\circ}_{corr}$ are the corrosion rates of steel samples in the absence and presence of St63Gly37, respectively.

A potentiostat (Amel 549) and linear sweep generator (Amel 567) were used to record the current–voltage curves. The scan rate was 1 V min$^{-1}$. The reference electrode was a saturated calomel electrode (SCE); the counter electrode was the platinum electrode. The working electrode was polarized at 800 mV for 10 min before recording the cathodic curves. For the anodic curves, the potential of the electrode was swept from its open circuit value after 30 min. A Voltalab PGZ-100 electrochemical system was used for the determination of the electrochemical impedance spectroscopy (EIS) at $E_{corr}$ after immersion in solution. After determination of the steady-state current at a given potential, sine wave voltage (10 mV) peak to peak, at frequencies between 100 kHz and 10 mHz, was superimposed on the rest potential. The measurements performed at rest potentials after 30 min

of exposure were automatically controlled by computer programs. EIS diagrams are detailed in the Nyquist representations.

## 3. Results

### *3.1. St63Gly37 Characterization*

The grafting of the glycerin may take place preferentially on the 7-position of the starch. It could also occur on positions 8 or 9 of the starch (Figure 2). On the other hand, glycerin can be grafted via its OH-13 or OH-14 functions. For the characterization of our products, we used $^1$H NMR and Fourier transform infrared (FT-IR) spectroscopy techniques.

### 3.1.1. $^1$H NMR Characterization

The $^1$H NMR spectrum of the starch grafted with glycerin St63Gly37 was recorded on a Brucker spectrometer of 400 MHz and dissolved in DMSO as solvent. Results are displayed in Figure 3.

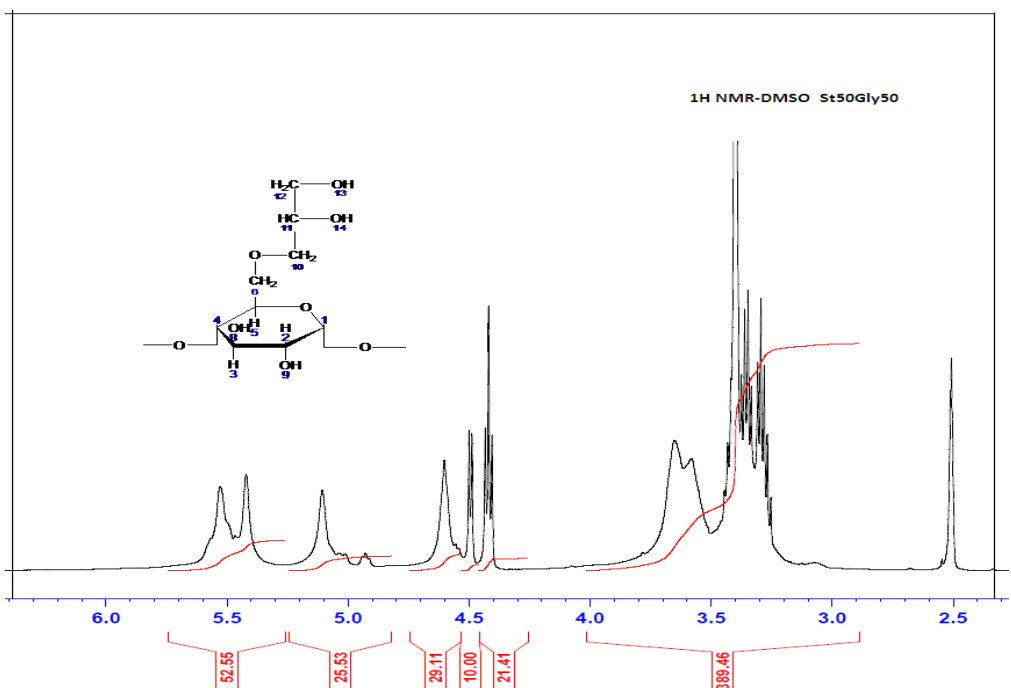

**Figure 3.** $^1$H NMR Spectrum of St63Gly37.

### 3.1.2. FT-IR Characterization

The FT-IR technique is largely used for the characterization of starch as a natural polymer [20]. Examination of the FT-IR spectrum confirms the grafting of glycerin on the starch, by the $CH_2$ bands and the characteristic OH bands of glycerin. Figure 4 shows the FT-IR spectra of the native starch in comparison with the St63Gly37 copolymer.

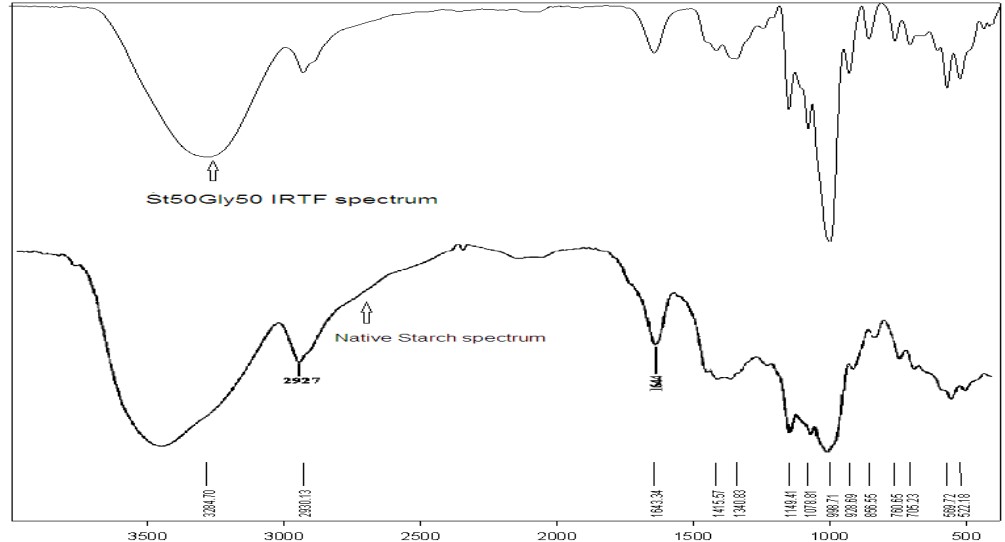

**Figure 4.** FT-IR spectra of the St63Gly37 copolymer (top) and native starch (bottom).

## 3.2. Weight Loss Measurements

### 3.2.1. Effect of Inhibitor Concentration

The results of C-Mn steel in HCl solution with different concentrations of St63Gly37 at 25 °C, using weight loss measurements, are reported in Table 2.

**Table 2.** Inhibition efficiency ($E_w$%) of C-Mn steel in 1 M HCl solution at different concentrations of St63Gly37, measured by weight loss at 25 °C.

| Inhibitor Concentration (mg L$^{-1}$) | $W_{corr}$ (mgcm$^{-2}$h$^{-1}$) | $E_w$ (%) |
|:---:|:---:|:---:|
| 0 | 0.661 | - |
| 5 | 0.471 | 29 |
| 10 | 0.404 | 39 |
| 50 | 0.330 | 50 |
| 100 | 0.225 | 66 |
| 200 | 0.113 | 83 |
| 300 | 0.038 | 94 |

$E_w$%—Inhibition efficiency, $W_{corr}$—corrosion rates.

### 3.2.2. Effect of Temperature

In order to study the effect of temperature on corrosion inhibition of C-Mn steel in HCl solution after two hours at different bio-copolymer concentrations, weight loss studies were carried out in a temperature range from 25 to 50 °C.

The variation of corrosion rate ($W_{corr}$) and inhibition efficiency $E_w$ (%) with the temperature for different concentrations of St63Gly37 bio-copolymer are displayed in Figures 5 and 6, respectively.

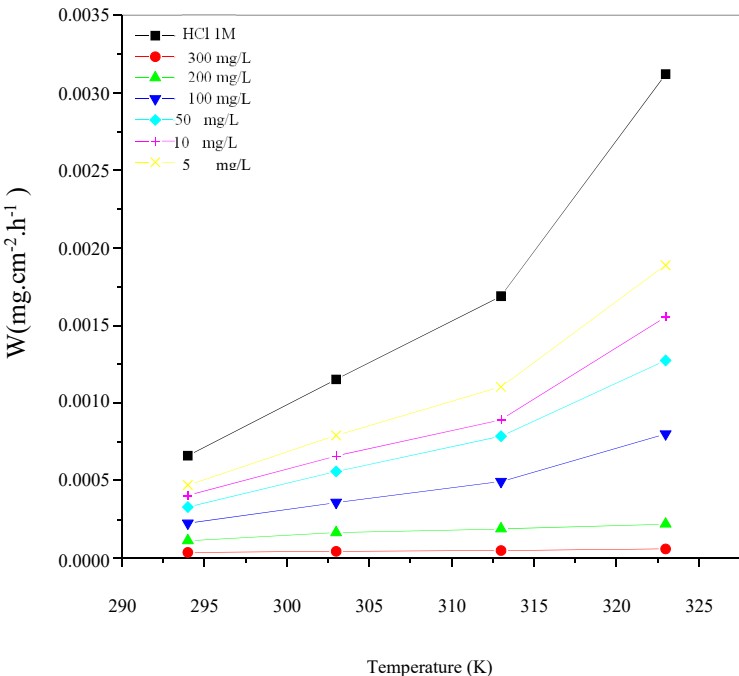

**Figure 5.** Variation of corrosion rate ($W_{corr}$) as a function of temperature for different concentrations of St63Gly37 bio-copolymer.

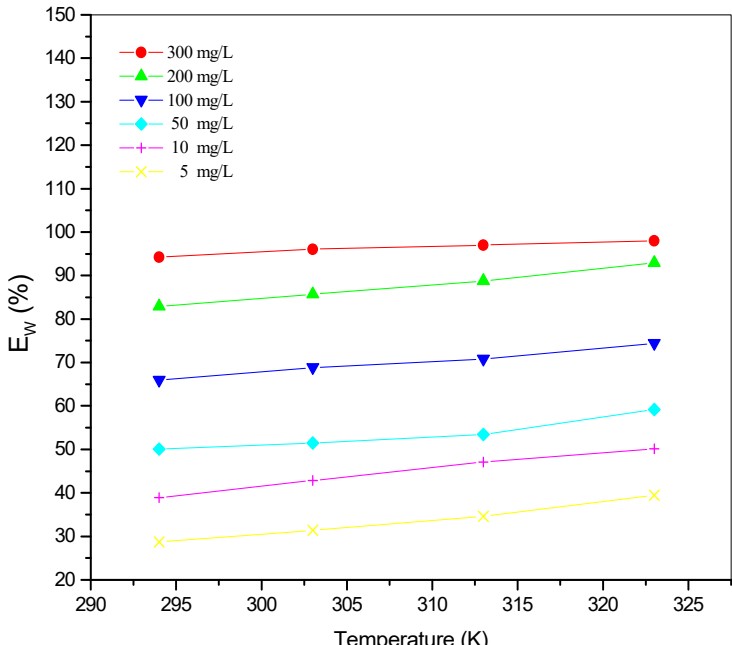

**Figure 6.** Variation of inhibition efficiency $E_w$ (%) as a function of temperature for different concentrations of St63Gly37 bio-copolymer.

### 3.3. Thermodynamic and Kinetic Parameters

Thermodynamic and kinetic parameters, such as activation energy $E_a$, enthalpy, and entropy of adsorption of St63Gly37 on steel, were calculated building the Arrhenius plot. Figure 7 presents the Arrhenius plots of corrosion rate logarithm vs. 1000/T related to blank solution and bio-copolymer solution.

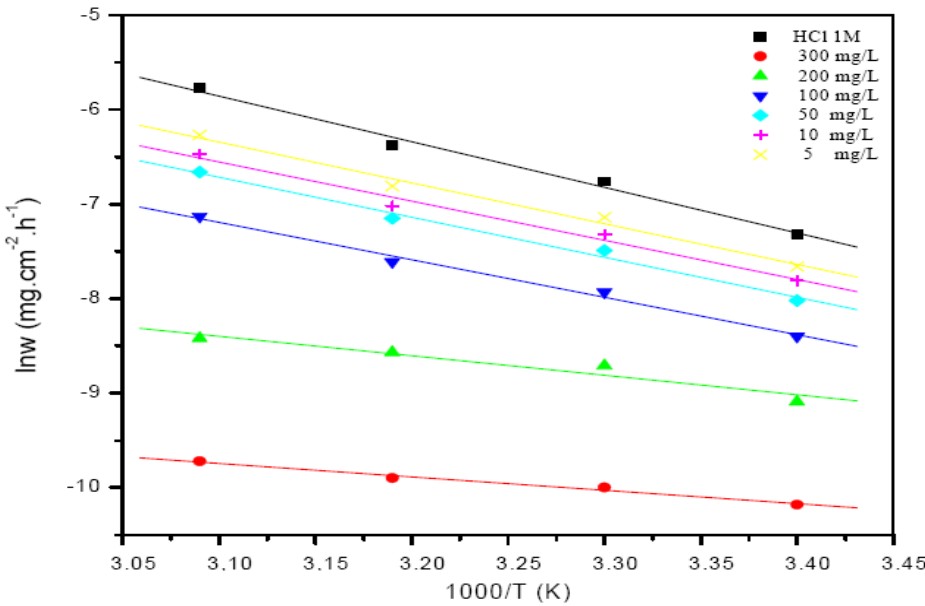

**Figure 7.** Arrhenius plots of the corrosion rate for both the blank solution and the solution of bio-copolymer.

The activation parameters for the corrosion process can be regarded as an Arrhenius-type process according to the following Equations (2) and (3):

$$\ln(W_{corr}) = \frac{-E_a}{RT} + A. \tag{2}$$

$$\ln(W'_{corr}) = \frac{-E'_a}{RT} + A. \tag{3}$$

$E_a$ and $E'_a$ are the apparent activation energies with and without the bio-copolymer, respectively. $T$ is the absolute temperature, $A$ is a constant, and $R$ is the universal gas constant. $W_{corr}$ and $W'_{corr}$ are the steel corrosion rates in the absence and presence of the bio copolymer inhibitor, respectively.

### 3.4. Adsorption Isotherms

Figure 8 shows the results of adsorption isotherms. Linear plots were obtained in the studied temperature range.

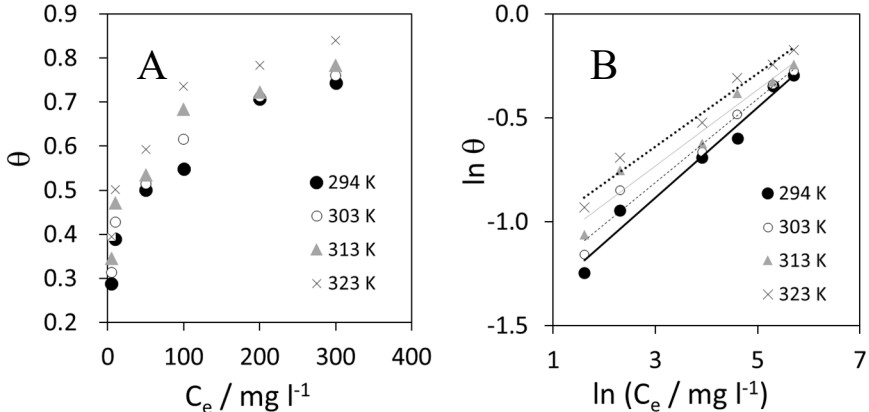

**Figure 8.** (**A**) Freundlich adsorption plot at different temperatures in the studied range; and (**B**) linearization of Freundlich isotherm.

The equilibrium constant of adsorption K is related to the standard free energy $\Delta G_{ads}$ [21]. $\Delta G_{ads}$ values at different temperatures can be calculated by Equation (4):

$$K = \frac{1}{55.5} \exp(\frac{-\Delta G_{ads}}{RT}), \tag{4}$$

where 55.5 represents the concentration of water in solution expressed in mol $L^{-1}$.

*3.5. Electrochemical Tests*

3.5.1. Polarizations

Current–potential plots resulting from cathodic and anodic polarization curves of steel in 1 M HCl in the presence of the studied bio-copolymer at various concentrations were recorded. The Tafel plots, recorded with a cathodic-to-anodic polarization of the system are shown in Figure 9.

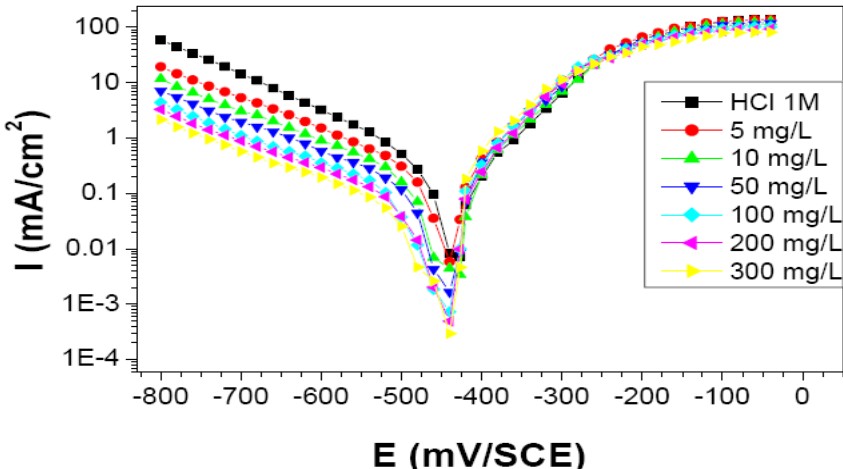

**Figure 9.** Cathodic and anodic polarization curves of C-Mn steel in 1 M HCl at different concentrations of St63Gly37. Scan rate 1 V min$^{-1}$.

The electrochemical parameters and the inhibition efficiencies ($E_I$), determined by the following Equation (5), are presented in Table 3:

$$E_I\% = \left(1 - \frac{I_{corr}}{I_{corr}^\circ}\right) \cdot 100, \tag{5}$$

where $i_{corr}$ and $i_{corr}^\circ$ are the corrosion current density values with and without St63Gly37 bio-copolymer inhibitor, respectively, determined by extrapolation of the cathodicbranch of the Tafel plot.

**Table 3.** Polarization parameters ($E_{corr}$ and $I_{corr}$) values and inhibition efficiencies of C-Mn steel corrosion in 1 M HCl at different concentrations of St63Gly37 bio-copolymer at 298 K.

| Inhibitor Concentration (mg L$^{-1}$) | $E_{corr}$ (mV vs. SCE) | $\beta_c$ (Vdec$^{-1}$) | $i_{corr}$ (µAcm$^{-2}$) | $E_I$ (%) |
|---|---|---|---|---|
| 0 | −439 | 0.155 | 304 | - |
| 5 | −440 | 0.180 | 223 | 27 |
| 10 | −444 | 0.185 | 192 | 37 |
| 50 | −455 | 0.190 | 157 | 48 |
| 100 | −458 | 0.191 | 113 | 63 |
| 200 | −459 | 0.187 | 56 | 82 |
| 300 | −460 | 0.190 | 27 | 91 |

*$E_{corr}$—corrosion potential, $i_{corr}$—corrosion current density, $E_I$—inhibition efficiencies, $\beta_c$—Tafel slope constant.*

### 3.5.2. Electrochemical Impedance Spectroscopy

A study of C-Mn steel corrosion behavior in 1 M HCl solution with and without St63Gly37 at 298 K was studied by EIS after an immersion time of 30 min. The purpose was to compare and complete the results obtained by the previous weight loss and polarization methods [22]. Nyquist diagrams obtained in the presence of various concentrations of bio-copolymer are shown in Figure 10. The deduced impedance parameters, as charge transfer resistance $R_t$ ($\Omega$ cm$^2$), double-layer capacitance $C_{dl}$ ($\mu$Fcm$^{-2}$), and inhibition efficiency ($E_{Rt}$%), are shown in Table 4.

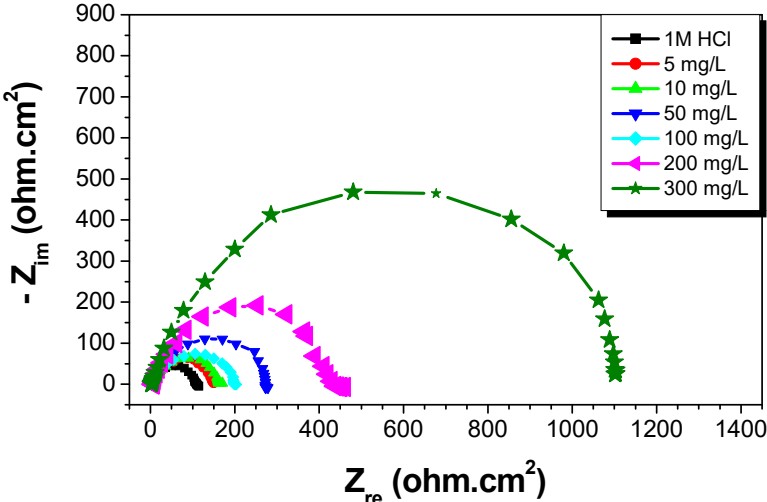

**Figure 10.** Nyquist plots for C-Mn steel in 1 M HCl at different concentrations of St63Gly37 bio-copolymer (298 K).

**Table 4.** Impedance parameters for corrosion of C-Mn steel in 1 M HCl at different concentrations of St63Gly37 at 298 K.

| St63Gly37 (mg L$^{-1}$) | $R_t$ ($\Omega$cm$^2$) | $f_{max}$ (Hz) | $C_{dl}$ ($\mu$Fcm$^{-2}$) | $E_{Rt}$ (%) |
|---|---|---|---|---|
| 0 | 110 | 16.32 | 88.7 | - |
| 5 | 146 | 12.88 | 84.6 | 24 |
| 10 | 167 | 12.81 | 74.4 | 34 |
| 50 | 200 | 11.39 | 69.9 | 45 |
| 100 | 275 | 9.08 | 63.8 | 60 |
| 200 | 458 | 6.52 | 53.3 | 76 |
| 300 | 1100 | 4.57 | 31.6 | 90 |

$R_t$—charge transfer resistance, $f_{max}$—maximum frequency, $C_{dl}$—double-layer capacitance, $E_{Rt}$—inhibition efficiency from the charge transfer resistance.

The charge transfer resistance values were qualitatively estimated from the difference in impedance at lower and higher frequencies [23]. The double-layer capacitance was obtained at the frequency $f_m$, at which the imaginary component of the impedance is maximal ($-Z_{i,max}$) according the following Equation (6):

$$C_{dl} = \frac{1}{2\pi f_m \cdot Rt}. \tag{6}$$

The inhibition efficiency from the charge transfer resistance was calculated by the following Equation (7):

$$E_{Rt}(\%) = \frac{R'_t - Rt}{R'_t} \times 100, \tag{7}$$

where $R'_t$ and $R_t$ are the charge transfer resistances with and without the St63Gly37 bio-copolymer as a corrosion inhibitor, respectively.

## 4. Discussions

### 4.1. St63Gly37 Synthetisis

The $^1$H NMR spectrum interpretation of the product St63Gly37 is carried out by comparison with the peaks of the native starch found in the literature [20]. The spectrum of native starch is shown in Table 5. The glycerin has two chemical shifts at 4.3 ppm corresponding to the proton of the CH, and at 4.4 ppm corresponding to the protons of $CH_2$.

**Table 5.** Chemical shifts of native starch [20].

| Chemical Shifts δ (ppm) | Attributions |
|:---:|:---:|
| 3.6 | H 2,3,4,5,6 |
| 4.6 | $H_7$ |
| 5.1 | $H_1$ |
| 5.6 | $H_8$, $H_9$ |

Chemical shifts of native starch and St63Gly37 bio-copolymer are comparable. The starch and glycerin signals are also found together, thus proving the success of the grafting reaction. The glycerin OH signals can be confused with those of the starch OHs ($H_8$, $H_9$).

After concluding that glycerin is well grafted on starch, the question that still remains is how this grafting takes place. Glycerin grafting may occur preferentially on the 7-position of the starch. It could also occur on positions 8 or 9 of the starch (see Figure 2). On the other hand, glycerin can be grafted via its OH-13 or OH-14 functions. Indeed, the possible reacting OH hydrogens are: $H_7$, $H_8$ and $H_9$. For the positions $H_8$ and $H_9$, they are still visible on the $^1$H NMR spectrum displayed in Figure 3 and Table 3. We can; therefore, conclude that $H_7$ reacted to give modified patterns according to the structure shown in Figure 2. The presence of the peaks at chemical shifts of 4.4 ppm (triplet) and 4.5 ppm (doublet) correspond, respectively, to the CH and $CH_2$ of the structure.

The $^1$H NMR spectrum also enables the calculation of relative amounts of starch and glycerin in St63Gly37. The copolymer contains 63% starch and 37% glycerin. The chemical shifts are represented in the following Table 6.

**Table 6.** Chemical shifts δ (ppm) of St63Gly37.

| Chemical Shifts δ (ppm) | Attributions |
|:---:|:---:|
| 3.2–3.7 | H 2,3,4,5,6 |
| 4.4 | H 11 |
| 4.5 | H 10 |
| 4.6 | H 13, 12 |
| 5.1 | H 1 |
| 5.4–5.6 | H 8,9,14 |

The FT-IR technique is largely used for the characterization of starch as a natural polymer [20]. Examination of the FT-IR spectrum confirms the grafting of glycerin on the starch, by the $CH_2$ bands and the characteristic OH bands of glycerin, as reported in Table 7. Figure 4 shows the FT-IR spectra of the native starch in comparison with the St63Gly37 copolymer. It is visible that the C–O–C bond bands change frequency because of the effect of the proportion of glycerin in the copolymer. The two spectra are almost of the same nature. The assignment of the different vibration bands of the native starch have been previously reported [20]. The comparison of the two FT-IR spectra shows a slight displacement of the elongation bands.

**Table 7.** Attribution of the different vibration bands of St63Gly37.

| Wavenumber: $\upsilon$ (cm$^{-1}$) | Attribution | Nature |
|---|---|---|
| below 600 | Vibration of the polysaccharide backbone | |
| 998.71–1000.10 | Antisymmetric C–O–C bond | Elongation |
| 1078.59 | Antisymmetric C–O–C bond | Elongation |
| 1149.84 | Antisymmetric C–O–C bond | Elongation |
| 1362.59–1359.95 | –OH in the plan | Deformation |
| 1416.76–1415.06 | C–H bond | Deformation |
| 1641.98–1644.62 | H–O–H vibration of adsorbed water | Deformation |
| 2929.18–2931.80 | C–H and CH2 of a polysaccharide | Elongation |
| 3274.32–3284.70 | OH associated | Elongation |

### 4.2. EffectOfinhibitor Concentration

The results of C-Mn steel in HCl solution with different concentrations of St63Gly37 at 25 °C, using weight loss measurements, are reported in Table 8.

**Table 8.** Data from the literature correlated to the type, concentration, and maximum inhibition efficiency ($E_{w,\max}$) obtained at 25 °C in HCl for Carbon-steel.

| Type of Inhibitor | Inhibitor Concentration (mg L$^{-1}$) | $E_{w,\max}$ (%) | Ref. |
|---|---|---|---|
| Glycerin-grafted starch | 300 | 94 | This work |
| *Ochrosiaoppositifolia* extract | 25 | 94 | [24] |
| Olive leavesextract | 900 | 91 | [25] |
| 2-amino-4-methylpentanoicacid | 7200 | 87.46 | [26] |
| L-tryptophan | 2000 | 90.8 | [27] |
| Gallicacid | 1000 | 59.64 | [28] |
| 3,5-bis(n-aminophenyl)-4-amino-1,2,4-triazole | 300 | 99 | [29] |
| Vanillin | 1520 | 86.1 | [30] |

Results showed that the inhibition efficiency calculated according to Equation (1) increases with increasing inhibitor concentrations. The highest concentration of inhibitor, equal to 300 mg L$^{-1}$, corres ponded to a maximum efficiency of 94%. For comparison, in Table 8, some data from the literature are reported. The cited papers deal with corrosion inhibition in a HCl environment by some investigated bio-based or synthetic compounds. A comparison of reported data demonstrates how the inhibition efficiency varies depending on the species involved in the inhibition mechanism. Moreover, it is also noticeable that the quantity needed to achieve a satisfying corrosion rate control strongly depends on the selected compounds. In a few cases, reported in Table 8, the used quantities are even excessive, losing the reasonable meaning of "corrosion inhibitors" as "compounds added in low amount to the aggressive environment". The concentration of 300 ppm used in this work was not increased more for the purpose of keeping the inhibitor amount in a logical range.

### 4.3. Effect of Temperature

The variation of corrosion rate ($W_{corr}$) and inhibition efficiency $E_w$ (%) in the temperature range 25–50 °C for different concentrations of St63Gly37 bio-copolymer, obtained by weight loss studies, are displayed in Figures 5 and 6.

Results showed that the inhibition efficiency increases with increasing temperature. As expected, corrosion process and inhibition efficiency are significantly dependent on the temperature [31–33]. Optimum temperature was found equal to 323 K, with a maximum efficiency of 98.07% with inhibitor concentration of 300 mg L$^{-1}$.

In the absence of corrosion inhibitor (blank solution), corrosion rate increases with increasing temperature, but when St63Gly37 bio-copolymeris added, the dissolution of C-Mn steel is widely

retarded. These results indicate that the corrosion inhibition mechanism might be more complex than a simple physisorption process on the steel surface. The values of inhibition efficiency, obtained using the weight loss method in the experimented temperature range, show that higher temperatures might favor the inhibitor sorption onto the steel surface. This might be explained in terms of chemisorption of polymer on the steel surface. In fact, in case of chemisorption, the extent of adsorption increases with rise in temperature, as reported in a previous work [34].

### 4.4. Thermodynamic and Kinetic Parameters

Results showed that the corrosion process for C-Mn steel increases more rapidly according to the temperature in the absence of inhibitor, rather than in its presence. This result confirms that the inhibitor acts as an efficient corrosion inhibitor in the range of temperatures studied.

Enthalpy and entropy of the corrosion process may be evaluated from the effect of temperature by an alternative formulation of transition state, as displayed in the following Equation (8) [35].

$$W = \frac{RT}{Nh} \exp\left(\frac{\Delta S^\circ_a}{R}\right) \exp\left(-\frac{\Delta H^\circ_a}{RT}\right), \tag{8}$$

where $h$ is Plank's constant, $N$ is Avogadro number, and $\Delta S^\circ_a$ and $\Delta H^\circ_a$ are the entropy and enthalpy of activation, respectively. Table 9 presents the calculated values of $E_a$, $\Delta S^\circ_a$, and $\Delta H^\circ_a$ in inhibited and uninhibited corrosive solutions. It is observed that the activation energy value is higher in the presence of the bio-copolymer inhibitor than in the uninhibited solution. The obtained activation energy value of the corrosion process in the inhibitor's presence, compared to its absence, can be attributed to its sorption. It is the result of electrostatic attraction between charged metal surface and charged species in solution and/or chemical interaction between polymer and metal.

**Table 9.** Calculated parameters at different concentrations of the bio-copolymer.

| Inhibitor Concentration (mg L$^{-1}$) | $E_a$ (kJmol$^{-1}$) | $\Delta H_a$ (kJmol$^{-1}$) | $\Delta S_a$ (Jmol$^{-1}$) |
|---|---|---|---|
| 0 | 40.9 | 38.5 | −36.7 |
| 5 | 36.6 | 34.1 | −37.1 |
| 10 | 35.7 | 33.3 | −37.2 |
| 50 | 35.3 | 32.9 | −37.4 |
| 100 | 26.5 | 24.0 | −37.5 |
| 200 | 27.5 | 25.0 | −38.0 |
| 300 | 28.8 | 26.3 | −38.1 |

($E_a$—activation energy, $\Delta S^\circ_a$ and $\Delta H^\circ_a$—the entropy and enthalpy of activation).

The values of $\Delta H^\circ_a$ are reported in Table 9. The positive sign of the reflects the endothermic nature of the steel dissolution process and values vary in the same way with inhibitor concentration and acid solutions [36].

On the other hand, values of are more positive in the uninhibited solutions and decrease by increasing the inhibitor concentration. Large and negative values of entropies imply that the activated complex in the rate-determining step represents an association rather than a dissociation step, meaning that a decrease in disordering takes place on going from reactants to the activated complex. A similar observation has been reported in the literature [37].

One can notice that $E_a$ and $\Delta H^\circ_a$ values vary in the same way (Figure 11). This result allows for the verification of the known thermodynamic relationship between the $E_a$ and $\Delta H^\circ_a$, as shown in Equation (9).

$$\Delta H^\circ_a = E_a - RT \tag{9}$$

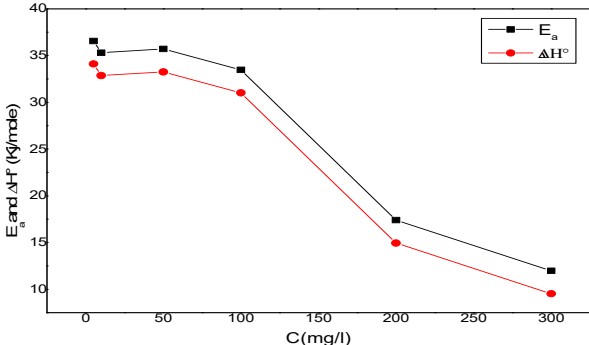

**Figure 11.** $E_a$ and $\Delta H°$ variation depending on the corrosion inhibitor concentration.

### 4.5. Adsorption Isotherm

As reported above, data indicate that a complex interaction mechanism takes place between the steel and the inhibitor, and a chemisorption might occur. Given this awareness, the analysis, reported in this paragraph, follows anyway the literature approach of applying isotherm adsorption models to verify the most experimental and realistic data. Adsorption isotherms are in fact used to understand the mechanism of metal–inhibitor interaction. The most frequently used isotherms are Langmuir, Frumkin, Temkin, and Parson [38,39]. The type of adsorption isotherm provides information about the interaction among both the adsorbed molecules themselves and their interactions with the metal surface. The most widely used isotherm employs the Langmuir model, whose primary assumptions are: (i) Adsorbate molecules attach to the active sites of the adsorbent surface, (ii) the Langmuir equation assumes that adsorption is monolayer, and (iii) all the sites on the solid surface are equal in size and shape and have equal affinity for adsorbate molecules [40]. However, the last two conditions are hard to fulfill in the corrosion studies, and this is the main weak point in terms of using the Langmuir model, as already evidenced by the literature [8]. Many practical cases, in fact, cannot be described the Langmuir model. The more complex adsorption models take into consideration factors such as the surface heterogeneity and the presence of areas having different adsorption energy or interactions between the adsorbed molecules.

The Freundlich isotherm is empirical and very widely used to describe the adsorption characteristics when the energy of adsorption on a homogeneous surface is independent of surface coverage. Its linearized form is described by the following Equation (10):

$$\ln \theta = \ln K + \frac{1}{n} \ln C_e \tag{10}$$

where $\theta$ is the surface coverage (calculated as $E_w/100$), $C_e$ is the adsorbate concentration in solution at equilibrium (in mg L$^{-1}$), Kis the equilibrium constant, and $1/n$ is a measure of intensity of adsorption. If the plotln $\theta$ vs. ln $C_e$ displays a linear trend, it means that the adsorption process can be described by the Freundlich model [41].

If $\Delta G_{ads}$ is lessthan $-10.40$ kJmol$^{-1}$, it can be inferred that the inhibitor interacts on the C-Mn steel surface by electrostatic effect.

The negative values of $\Delta G°_{ads}$, displayed in Table 8, confirmed the spontaneity of the process and stability of the adsorbed layer on the steel surface [42,43]. The obtained values of $\Delta G°_{ads}$ show the dependence of $\Delta G°_{ads}$ on temperature (Table 10), indicating a strong interaction between the bio-polymer molecules and the metal surface.

Thermodynamically, $\Delta G°_{ads}$ is related to the standard enthalpy and entropy of the adsorption process, $\Delta H°_{ads}$ and $\Delta S°_{ads}$, respectively, via the Equation (11):

$$\Delta G°_{ads} = \Delta H°_{ads} - T\Delta S°_{ads} \tag{11}$$

The positive value of $\Delta H°$ suggests an endothermic nature of the metal dissolution process in the presence of the inhibitor. The positive value of $\Delta S°$ indicates that the adsorption process is accompanied by an increase in entropy, which is the driving force for the adsorption of the inhibitor onto the metal surface [31].

**Table 10.** Variation of the thermodynamic parameters according to the Langmuir isotherm at the studied temperatures.

| T (K) | $K_{ads}$ | $\Delta G°_{ads}$ (kJmol$^{-1}$) | $\Delta H°_{ads}$ (kJ mol$^{-1}$) | $\Delta S°_{ads}$ (Jmol$^{-1}$K$^{-1}$) |
|-------|-----------|-----------------------------------|------------------------------------|------------------------------------------|
| 294 | 0.223 | −6.15 | | |
| 303 | 0.244 | −6.56 | | |
| 313 | 0.278 | −7.12 | 9.22 | 52.2 |
| 323 | 0.312 | −7.65 | | |

$K_{ads}$—adsorption parmeter, $\Delta G°_{ads}$—adsorption enthalpy, $\Delta H°_{ads}$—enthalpy, $\Delta S°_{ads}$—entropy of the adsorption process.

### 4.6. ElectrochemicalResults

#### 4.6.1. Polarizations

The values reported in Table 3 showed results in agreement with what was obtained by the weight loss method: Corrosion current densities decrease when the concentration of the inhibitor increases. Maximum value of inhibition efficiency of 91% was obtained for inhibitor concentration of 300 mg L$^{-1}$.

The obtained polarization curves indicate that the addition of the bio-copolymer influences the kinetics (decrease in reaction rate) but does not modify the mechanism of the cathodic process [22].

From a thermodynamic point of view, at the acidic pH (1 MHCl) and $E_{corr}$ (ca. −450 mV vs. SCE) of the investigated system, both the oxygen reduction reaction and hydrogen evolution reaction could potentially take place as the iron Pourbaix diagram states [23].

As for the kinetic aspects related to the cathodic Tafel slope calculation, it is apparent that slope values are markedly different from the theoretical ones expected in the case of pure hydrogen evolution reaction. Tafel slope is in fact known to be 118, 39, or 29.5 mV/decade when the discharge reaction, electrochemical desorption reaction, or recombination reaction is rate-determining, respectively [44]. This might mean that more than one reaction is taking place in the considered potential range.

However, it is shown in Table 3 that Tafel slopes do not remarkably change with addition of different concentrations of inhibitor, thus confirming the hypothesis that the inhibitor does not significantly change the cathodic reaction mechanisms. Therefore, the use of the polarization curves as a tool for deriving quantitative parameters, such as inhibition efficiency, can be justified since it is based on relative calculations (using current densities values in the presence and absence of inhibitor) and does not imply an absolute evaluation.

Considering the anodic reaction, C-Mn steel corrodes in acid medium according to an electrochemical process by anodic dissolution of iron. This phenomenon might be accentuated if other parameters intervene, such as microbiological activity or mechanical stresses. The almost constant value of $E_{corr}$ at a first sight might give an indication that the inhibitor is of mixed type acting by adsorption mechanism. However, the cathodic branches trend, due to inhibitor concentration increase, and the overlap of all the anodic branches points to the inhibitor being a cathodic one. This is also in agreement with results obtained at different temperatures. In fact, data showed that the corrosion inhibition mechanism might be more complex than a simple physisorption process on the steel surface.

#### 4.6.2. Electrochemical Impedance Spectroscopy

According to results reported in Figure 10 and Table 4, the impedance diagrams show a semi-circle, indicating that the charge transfer process mainly controls the steel corrosion. By increasing the concentration of St63Gly37, the charge transfer resistance increases and double-layer capacitance

(*Cdl*) decreases. A minimum value of *Cdl b* of 31.64 μFcm$^{-2}$ was obtained for inhibitor concentration of 300 mg L$^{-1}$. The decrease in *Cdl* can be explained by the sorption of inhibitor compound on steel surface leading to the protection from the aggressiveness of HCl. The kinetics of the cathodic hydrogen evolution reaction is decreased and the chlorides are prevented from crossing the protective barrier [26,27]. The resulting inhibition efficiency ($E_{Rt}$) increases progressively when the concentration of the inhibitor increases. A maximum value of 90% was obtained for inhibitor concentration of 300 mg L$^{-1}$. These obtained values are in agreement with results obtained by weight loss and polarization method.

## 5. Conclusions

C-Mn steel in 1 M HCl solution was investigated to study the corrosion behavior in the temperature range 25–50 °C with and without a glycerin-grafted starch as a bio-copolymer working as a corrosion inhibitor.

Results obtained by the weight loss method showed that the inhibition efficiency increases by increasing the inhibitor concentration. Optimum concentrations of inhibitor equal to 300 mg L$^{-1}$ enabled a maximum efficiency of 94.25%.

The corrosion process and the inhibition efficiency were found to be significantly dependent on the temperature and concentration of inhibitor. The values of inhibition efficiency, obtained using the weight loss method in the experimented temperature range, show that higher temperatures might favor the inhibitor sorption onto the steel surface. Results indicated that the corrosion inhibition mechanism might be more complex than a simple physisorption process on the steel surface. This might be explained in terms of chemisorption of polymer on steel surface.

The obtained values of corrosion potential and corrosion current density, $E_{corr}$ and $i_{corr}$, obtained by potentiodynamic polarization, are in agreement with the weight loss method: The corrosion current densities decrease when the concentration of the inhibitor increases. An analysis of data indicates that the inhibitor might be of the cathodic type.

The decrease in double-layer capacitance values, obtained by EIS method, follows a decrease similar to that obtained for $i_{corr}$ by the polarization method. The decrease in $C_{dl}$ can be explained by the sorption of inhibitor compound on steel surface leading to the protection from the aggressiveness of HCl.

Results obtained in this study might be developed to supplement corrosion protection in real field applications, injecting the optimum quantities of corrosion inhibitor without any harmful effect on the natural environment and human health.

**Author Contributions:** Investigation, B.B.; Methodology, A.M.; Supervision, A.B.; Writing–original draft, S.L.; Writing–review & editing, L.T. and S.M.

**Funding:** This research was funded by Direction Générale de la recherché scientifiqueet du développement technologique, Algerian Ministry of Scientific Research.

**Acknowledgments:** The authors are very grateful to the Algerian Gas Transport TRC for their great help to make a part of this work by providing us with some steel samples. They also thank all researchers at the LAEPO Research Laboratory from Tlemcen University for their experimental contribution during the corrosion test, and the equipment support from the Algerian University and Society. They also appreciate the contribution of the Department of Environmental Science and Policy from Università degli Studi di Milano.

**Conflicts of Interest:** The authors declare no conflicts of interest. The funders had no role in the design of the study; in the collection, analyses, or interpretation of data; in the writing of the manuscript, or in the decision to publish the results.

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
