# Peer review of "Glycerin-Grafted Starch as Corrosion Inhibitor of C-Mn Steel in 1 M HCl solution"

_applsci, doi:10.3390/app9214684_

Round 1
Reviewer 1 Report
The publication presents a set of typical tests that are used to characterize corrosion inhibitors. The subject of "green inhibitors" is currently being widely developed.The authors presented the results of the study of the inhibitor, which was synthesized by them. The introductory part of the article could be slightly developed due to the very large number of publications appearing in the literature on "green inhibitors".The test results are consistent and clearly indicate an increase in the effectiveness of the inhibitor with increasing concentration. Based on the analysis of the corrosion potential, the authors conclude that the inhibitor works in an anode manner. The slight changes in potential with increasing concentration of the inhibitor actually suggest a mixed effect of the inhibitor. In fig. 11 an error related to the determination of inhibitor concentration crept in.
The highest concentration was inserted twice and probably some lower were omitted. It would also be useful to carry out impedance tests as the charge transfer resistance changes over a longer exposure. Formula (11) was incorrectly presented and described.In the form presented with the current description, the effectiveness values of the inhibitor would be negative.
Author Response
Dear Sir
The authors have been revised the manuscript according to the reviewer's comment. Please see the attachment
Professor A. Benmoussat

Reviewer 2 Report
This paper employed a bio-copolymer to protect C-Mn steel from corrosion. The results are sound and informative. I would suggest that the paper should be accepted after some English polishing.
Author Response
Dear Sir
The authors have been revised the manusript acording the reviewer's comments. Please see the attachment
Prof A. Benmoussat
